# Evaluation of Dysphagia and Inhalation Risk in Neurologically Impaired Children Using Esophageal High-Resolution Manometry with Swallowing Analysis

**DOI:** 10.3390/children9121987

**Published:** 2022-12-17

**Authors:** Anna Maria Caruso, Denisia Bommarito, Vincenza Girgenti, Glenda Amato, Adele Figuccia, Alessandra Casuccio, Annalisa Ferlisi, Rosaria Genuardi, Sabrina La Fata, Rosalia Mattei, Mario Pietro Marcello Milazzo, Maria Rita Di Pace

**Affiliations:** 1Pediatric Surgical Unit, Children’s Hospital ‘G. di Cristina’, ARNAS Civico, 4, 90127 Palermo, Italy; 2Pediatric Surgical Unit, Department Health Promotion, of Mother and Child Care, Internal Medicine and Medical Specialities, University of Palermo, 61, 90133 Palermo, Italy; 3Cystic Fibrosis and Respiratory Pediatric Unit, Children’s Hospital ‘G. di Cristina’, ARNAS Civico, 4, 90127 Palermo, Italy; 4Pediatric Anestesiologit Intensive Unit Care Children’s Hospital ‘G. di Cristina’, ARNAS Civico, 4, 90127 Palermo, Italy; 5Medical Direction, Pediatric Nutritional Service, Children’s Hospital ‘G. di Cristina’, ARNAS Civico, 4, 90127 Palermo, Italy

**Keywords:** neurological children, dysphagia, inhalation, high-resolution manometry

## Abstract

Background: Dysphagia in neurologically impaired children is associated with feeding difficulties, malnutrition and aspiration pneumonia. Esophageal high-resolution manometry (HRM) has been used in the diagnosis of motility disorders affecting the swallowing process. The aim of this study was to analyze swallowing functions in NI children by using HRM in order to establish swallow parameters identifying inhalation risk. Methods: Twenty-five NI children with cerebral palsy were submitted to esophageal HRM with UES analysis, comparing the results with non-NI children. The following parameters were evaluated: maximum pressure and duration of contraction of the velopharynx (VP) and tongue base (TB), and maximal, minimal, resting pressure and relaxation duration of the upper esophageal sphincter (UES). Results: pVP max, pTB max, pUES max and resting pressure were lower, while *p* UES minimal was higher and relaxation duration was shorter in NI children vs. the control group. Predictive values of inhalation risk were evaluated. Conclusions: This study evaluates inhalation risk in NI children using HRM to study UES function. Our results confirm the alterations described in NI children: insufficient contraction and clearing force for bolus transmission through the pharynx and incomplete UES relaxation can predispose to pharyngeal residues and inhalation independently of swallowing because of lower values of UES resting.

## 1. Introduction

The increasing survival of children with severe central nervous system damage has created a major challenge for medical care. Cerebral palsy (CP) is defined as a group of permanent disorders of development, movement and posture that cause activity limitation and are attributed to nonprogressive disturbances in the developing fetal or infant brain. CP is considered as the major subgroup of neurological impairment (NI) [1,2,3,4]. Although the primary issues for patients with developmental disabilities are physical and mental incapacities, several clinical reports indicate that brain damage may also result in significant gastrointestinal dysfunction, most notably oral motor function and motility [5,6,7,8]. NI children typically report feeding difficulties, extended feeding times, malnutrition and/or a history of aspiration pneumonia; before the development of severe and complicated undernutrition, ESPGHAN WG recommends to start with enteral tube feeding with percutaneous endoscopic gastrostomy (PEG) when oral feeding is deemed to be ineffective and unsafe [9,10]. Swallowing is deemed unsafe in cases of both a history of aspiration pneumonia (antibiotics or hospital admission for chest infection) associated with objective evidence of aspiration or penetration on contrast video fluoroscopy (VFSS), or fiberoptic endoscopic swallowing evaluation (FEES) [9,10,11]. VFSS remains a key diagnostic exam for the assessment of dysphagia in children, but it has several limitations, especially in NI patients [12,13,14,15,16]. Esophageal high-resolution manometry (HRM) is mainly applied in the evaluation of esophageal disorders with dysmotility, allowing the direct and fast evaluation of the swallowing process without using radiation or contrast materials; however, only a few studies report the use of pharyngeal HRM for patients with dysphagia, taking into consideration the previously demonstrated relationship between VFSS and HRM parameters [17,18,19,20,21]. A study between healthy and dysphagic non-NI adult subjects was performed by Parks, who studied the efficacy of HRM in detecting pharyngeal motility and function, analyzing sensitivity, specificity and the predictive values of manometric parameters, and demonstrating significant differences between the two groups [22]. As far as we know, similar studies in NI children evaluating dysphagia and inhalation risk are not described in the literature. The aim of this study was to compare swallowing function in two groups (NI and non-NI children) using HRM, thus establishing threshold values for HRM swallow parameters and determining their sensitivity, specificity and predictive values for inhalation risk. 

## 2. Materials and Methods

### 2.1. Patients

A prospective study was conducted in our center from January 2020 to June 2022. Twenty-five (25) NI children with cerebral palsy, up to 18 years of age, followed for impaired swallowing and undernutrition, are included in this study. We excluded patients with neuromuscular disease.

First, PEG placement was indicated in 13 pts because of a history of recurrent pneumonia with hospitalization associated with a severe status of malnutrition for 9 of them; 12 pts were already carrying a PEG or JPEG (jejunal PEG) with exclusive tube enteral feeding. Despite their better nutritional status, however, they still showed recurrent pulmonary events, suspected to be due to inhalation/aspiration. Among the 12 pts already on enteral feeding, 7 had a diagnosis of pharyngeal dysphagia based on VFSS data (bolus retention in pharyngeal area, penetration and aspiration in the airway), whereas the remaining 5 pts had a negative VFSS and received PEG because of severe malnutrition. 

All 25 NI pts underwent a fiberoptic endoscopic swallowing evaluation (FEES), the results of which were clearly altered in 18 pts only. All 25 NI pts then underwent esophageal HRM with UES analysis; their results have been compared with a control group of 25 non-NI children: pediatric patients in follow-up at the digestive surgery clinic, with normal oral feeding, without any malformation, and undergoing esophageal manometry for different pathologies, such as gastroesophageal reflux, epigastric pain and caustic esophagitis. Patients with esophageal malformation or history of oropharyngeal surgery were excluded from study for both groups.

The primary outcome of this study was to use HRM to analyze UES function in children with neurological impairment; the second outcome was to evaluate any difference with children who are not neurologically impaired and establish predictive parameters of inhalation risk. 

The study protocol was approved by the Institutional Review Board of our center (n. protocol 17/2016, administration number 353). This study was conducted in accordance with the 1975 Helsinki Declaration, as revised in 2008. All participants and/or their responsible guardians gave written consent after being informed about the nature of the study.

### 2.2. HRM Procedure

This study was conducted using ManoScan^®^ esophageal high-resolution manometry (Medtronic, Dublin, Ireland). Data acquisition, display and analysis are based on ManoView Software according to previously established methodologies [20]. 

A solid-state high-resolution manometric assembly (outer diameter 4.2 mm) with 36 circumferential sensors spaced at 1 cm intervals was used. The HRM procedure was conducted with a semi-seated patient, after at least 3 h of fasting (both oral and tube meals). The HRM catheter position was confirmed when the esophageal gastric junction was highlighted, as in standard esophageal manometry; analysis of the UES zone was performed at the end of the examination, reviewing the track with a zoom image of 8 s from the nasal cavity (VP) to the upper esophageal sphincter (UES).

During the exam, HRM swallow values were assessed with five boli of 5 mL semisolid meal to reduce the inhalation risk that would have occurred with a liquid; means for each of the HRM parameters were then calculated. We defined the following regions of interests (ROI) manometrically: velopharynx (VP), tongue base (TB) and upper esophageal sphincter (UES) (Figure 1).

VP coincides with the area of pressure change related to swallowing, proximal to the quiescence of nasal cavities, extending 2 cm downwards and corresponding to the soft palate (velum) and posterior pharynx. TB is a swallow-related pressure zone between VP and UES. The UES region is a stable high-pressure zone displayed immediately above low esophageal pressure that anatomically corresponds to the cricopharyngeal and inferior pharyngeal muscles. Several parameters, such as VP maximal pressure, VP duration, TB maximal pressure, TB duration, UES maximal pressure, UES minimal pressure, UES relaxation duration and UES resting pressure, were evaluated. The time of contraction, with respect to the baseline, corresponded to the interval between the onset and offset of pressure change. UES maximal and minimal pressure are the maximal and minimal UES pressure values, respectively, during UES relaxation in the swallowing phase. UES relaxation duration is the interval from the onset of contraction (starting from half the baseline) to the offset visualized as return to half-baseline [21,23]. UES resting pressure is defined as the maximal value occurring during the normal respiratory cycle (not swallowing phase) [17,24,25]. Means and standard deviations (SD) of each HRM parameter have been described.

### 2.3. Statistical Analysis 

Statistical analysis of quantitative and qualitative data, including descriptive statistics, has been performed for all items. The Shapiro–Wilk test was used to evaluate the normality of the distribution of the quantitative data. Continuous data have been expressed as mean ± SD unless otherwise specified. Variable differences between groups have been assessed by the independent Student’s *t*-test. To assess the predictive rate of different cutoff values of HRM parameters with regard of patient groups, a receiver operating characteristic (ROC) curve with calculations of area under the curve and 95% CI has been constructed, and sensitivity and specificity values have been calculated. Data were analyzed with IBM SPSS Software 22 version (IBM Corp., Armonk, NY, USA). All *p*-values were two-sided and *p* < 0.05 was considered statistically significant.

## 3. Results

Demographic data, with differences between the NI children and the control group, are reported in Table 1. The control group was significantly older than the NI group (*p* < 0.005); BMI in the NI group was lower than in the control group (*p* < 0.005); no differences were found regarding age and gender (*p* 0.181). 

Differences regarding HRM parameters between the two groups are summarized in Table 2. 

VP and TB maximal pressure, as well as UES resting pressure, were lower for the NI group with statistically significant differences; UES minimal pressure was higher in the NI group than in the control patients. Weaker differences were observed in TB and VP duration, as well as UES maximal pressure. Threshold values of significant impaired pharyngeal motility were identified with ROC analysis and are reported in Table 3.

For all parameters, threshold values have been identified with good values of AUC (area under ROC curve), specificity and sensibility; for TB maximal pressure and UES resting, the value of the ROC curve was 1.0 with 100% specificity and sensibility, which is the maximum obtainable value with ROC analysis. All ROC curves for each parameter are shown in Figure 2.

## 4. Discussion

Swallowing is the result of a semiautomatic motor action of the respiratory muscles, oropharynx and gastrointestinal tract that propels the bolus from the oral cavity to the stomach and protects the airway from food, liquids and other substances [26,27]. Dysphagia is defined as an impairment of this complex and integrated sensorimotor system that can occur in one or more of the oral, pharyngeal and esophageal phases. Related swallowing problems have been shown to affect up to 90% of NI children, being main contributors to malnutrition; dysphagia is related mainly to oral motor dysfunction with aspiration of liquid, without reactive clinical response (silent aspiration) [1,14,28]. Airway aspiration is a result of unsafe feeding or retrograde reflux events, and is very dangerous in NI children because it is associated with significant respiratory problems such as recurrent pneumonia, permanent lung damage and even death [29]. Hence, the early identification of swallowing disorders can prevent airway disease and serious complications too. At the moment, for children with neurodisability, VFSS remains one of the key investigations to assess dysphagia, with a number of studies supporting its utility in identifying discoordinate pharyngeal motility and silent aspiration, and in diagnostic work-up to guide effective feeding strategies. VFSS may also be used to assess other parameters relative to dysphagia and impaired feeding, such as reduced lip closure, inadequate bolus formation, residue in the oral cavity, delayed triggering of pharyngeal swallow, reduced larynx elevation, coating on the pharyngeal wall, delayed pharyngeal transit time and multiple swallow [12,13,14,15].

With VFSS, different patterns of dysphagia have been described [14]: in children with cerebral palsy, involvement of the motor tract is related to clinical aspects of spasticity, whereas that of basal ganglia or thalamus is related to dyskinetic features. Dysphagia in children with CP is related to oral motor problems and silent aspiration, whereas in children with neuromuscular disorders, several feeding problems, such as piecemeal deglutition and problems with semisolid and solid food, as opposed to thin liquid, are reported. These different patterns can be shown because with VFSS it is possible to study the complete swallow sequence: bolus formation and oral preparatory phase (loss of food out of the mouth caused by reduced lip closure, tongue thrust and reduced tongue control), oral transit phase (piecemeal deglutition, time of oral transport caused to reduced tongue movement and coordination), first pharyngeal phase (material in valleculae or pyriform sinuses preinitiation) and pharyngeal phase. During this phase, the following alterations can be evaluated: pharyngeal backflow caused by reduced velopharyngeal closure or incoordination of contraction; laryngeal penetration caused by reduced or delayed closure of airway entrance; aspiration before, during and after swallow; and post-swallow residues in valleculae and pyriform sinuses or posterior wall. In the last phase, (upper esophageal) post-swallow residue on/in the upper esophagus is evaluated. Compared with the neuromuscular disorder group, CP patients had significantly more loss of food out of the mouth, material in the valleculae already in preinitiation of pharyngeal swallow, pharyngonasal backflow, laryngeal penetration and aspiration with liquid. Inhalation in patients with CP is mainly related to altered tongue base retraction (necessary for swallow initiation), delayed or insufficient laryngeal elevation with reduced pharyngeal contraction and disturbed sensory information with consequent silent aspiration events.

Unfortunately, it has several limitations, mainly related to prolonged radiation exposure, allergies to contrast materials and the impossibility of application to bed-ridden patients. Data obtained from VFSS depend on an operator-dependent qualitative evaluation with only a few quantitative measurements [16,17].

Currently, esophageal HRM has been used in the assessment of motility disorders of the esophagus [17,18,19]; only a few studies have described the use of HRM in pharyngeal motility evaluation with UES analysis for patients with dysphagia using the strong correlation between VFSS and HRM parameters shown in studies with simultaneous evaluation [17,21,22,23,24,25]. Yoon et al. described the swallowing sequence, simultaneously considering both contrast in VFSS and manometric pressure changes in HRM using three swallow types (water, barium and yogurt) [21].

HRM has the advantage that it can be performed with a movable device that is easily carried to bed-ridden patients, and only minimal positioning of the patient is needed. Previously, conventional manometry, equipped with few sensors, showed limitations to simultaneously visualizing the esophagus from the pharynx to the stomach; to overcome this limitation, high-resolution manometry (HRM) was modified, adding total radial sensors located at 1 cm intervals or less on the tube (36 sensors) so that it could very accurately describe the swallow events visualized as generated pressure change with respect to the resting baseline [18,19,20]. The upper esophageal sphincter (UES) is functionally defined as an intraluminal high-pressure zone between the pharynx and the proximal esophagus. The UES relaxes transiently during swallowing so that the bolus can move in the esophagus; corresponding to this inhibition, right after the onset of deglutition, manometric recordings show a decrease in UES pressure [30,31].

Our results in NI children with HRM confirmed physio-pathologic alterations already described in the literature for children with CP [1,28,29,32]. In CP, different patterns of dysphagia compared with neuromuscular disease were described [14], and for these reasons we excluded patients with neuromuscular disease from our study.

In particular, we observed insufficient VP contraction and closure. In the initial pharyngeal phase of swallowing, the VP has the fundamental role of preventing nasal regurgitation and creating a tight seal with squeezing pressure between the velum and the pharyngeal wall; if the VP closure is insufficient during pharyngeal swallowing, nasal regurgitation of bolus material and reduced pressure in the upper pharyngeal portion can follow, leading to weak bolus propulsion force and remnant food material in the pharynx. A value of 155 mmHg appears to be an optimal threshold for identifying patients at risk of inhalation, with 100% sensitivity and specificity.

As for the TB max pressure, the lower value reported in our NI children could be associated with incomplete tongue base retraction during swallowing (as seen on VFSS studies too), and contraction and clearing force are insufficient to move bolus through the pharynx [32]; a shorter duration of TB and VP could be assumed to be a less effective contraction.

Because VP and TB are correlated with the pharyngeal contractile phase, measurement using HRM is very important in identifying pharyngeal dysfunction when these parameters are lower.

As for UES, its maximum pressure appears to be a less significant parameter, with fewer differences between the NI and control children; UES is not a pressure-generating structure and it appears important to observe its capacity to open and relax while swallowing rather than its contraction force. In fact, we found that UES minimal pressure and UES relaxation values were systematically higher in NI children; as the upper sphincter does not relax adequately, it leaves less time for the relaxation phase and causes pharyngeal residues.

As a last parameter, UES resting pressure was analyzed as the only value measured during the respiratory rather than the swallowing phase. This value in NI children is lower: a value of 65 mmHg appears predictive for inhalation risk, with 100% specificity and sensitivity. This could likely correspond to an inadequate resting contraction, protective of the airways, and therefore to predisposition to inhalation regardless of swallowing. This is a very important measurement, as it can be also performed in unconscious patients without risk of inhalation of the bolus.

An optimal threshold level with satisfying sensitivity and specificity has been identified for each HRM parameter, with differences compared to Parks’ results. These values could be used to support clinical practice in identifying patients at risk of inhalation, leading to an indication to exclusive enteral tube feeding, or used to evaluate patients with suspect aspiration events, especially in cases where the VFSS is negative.

The main limitations of this study were the small number of patients and the use of liquid boluses only, thus lacking analysis for solid bolus in order to evaluate the influence of the bolus consistency on UES function; we also considered only patients with CP, and thus did not evaluate any pathology-related differences in order to confirm with HRM the different dysphagia pattern described with VFSS.

## 5. Conclusions

In conclusion, to the best of our knowledge, this is the first study evaluating inhalation risk in NI children using HRM for UES function.

With this study, we want to propose a valid alternative to VFSS, especially in patients in which VFSS can be difficult to perform or when it is negative, despite a strong suspicion of inhalation (a false negative as a result of the employee operator report is a main limit of VFSS). The results obtained with HRM in our study are in agreement with those described up to now with VFSS in children with CP; different values (as numeric values) are partially in agreement with those described by Parks, but we do not have references regarding pediatric or neurological patients.

Although the data on HRM with UES analysis in children are limited, and our study was a preliminary analysis, nevertheless we think that HRM with appropriate equipment and expertise can be a valid alternative to VFSS, especially when it is difficult to differentiate UES dysfunction vs. pharyngeal dysmotility.

## Figures and Tables

**Figure 1 children-09-01987-f001:**
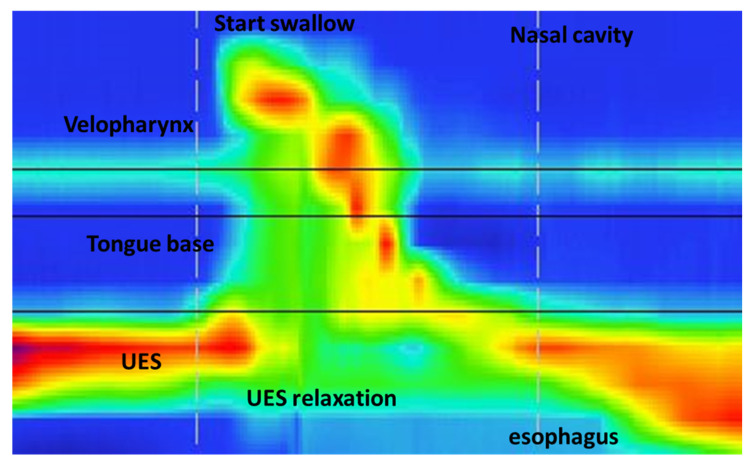
High-resolution manometry (HRM) with swallowing analysis. Regions of interest for HRM procedure between nasal cavity and esophagus. Velopharynx (VP), tongue base (TB) and UES are described. Start of swallow and UES relaxation are shown.

**Figure 2 children-09-01987-f002:**
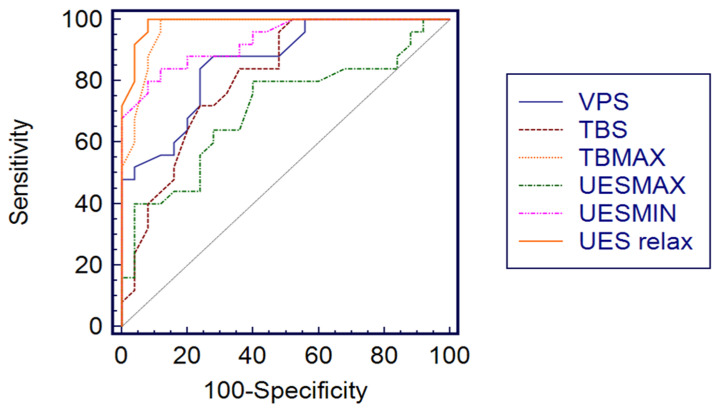
The threshold values of significant HRM parameters for identifying inhalation risk in ROC analyses. ROC: receiver operating curve; VPs: velopharynx duration (s); TBs: tongue base duration (s); TB max: tongue base maximal pressure; UES max: upper esophageal sphincter maximal pressure; UES min: upper esophageal sphincter minimal pressure; UES relax: upper esophageal sphincter relaxation.

**Table 1 children-09-01987-t001:** Patients’ characteristics with differences between the two groups.

Characteristic	NI Children (25)	Control Group (25)
Age (months)	105.48 ± 53.11	124.8 ± 47.25
Gender	12 M/13 F	10 M/15 F
BMI	13.36 (r 6–27)	20.45 (15–26)

NI: neurological impairment; BMI: body max index; M: male; F: female.

**Table 2 children-09-01987-t002:** Comparison of HRM parameters between NI children and control group.

HRM Parameters	NI Group Mean (± SD)	Control Group Mean (± SD)	*p*-Value
VP maximal pressure (mmHg)	120.96 ± 21.26	212.72 ± 42.46	<0.0005
VP duration (s)	0.61 ± 0.15	0.91± 0.27	<0.0005
TB maximal pressure (mmHg)	144.44 ± 53.43	243.84 ± 28.56	<0.0005
TB duration (s)	0.48 ± 0.11	0.68 ± 0.17	<0.0005
UES maximal pressure (mmHg)	251.08 ± 47.32	284.12 ± 36.01	0.008
UES minimal pressure (mmHg)	6.69 ± 2.79	1.49 ± 2.93	<0.0005
UES relaxation duration (s)	0.42 ± 0.10	0.74 ± 0.12	<0.0005
UES resting pressure (mmHg)	35.52 ± 20.24	103.32 ± 11.50	<0.0005

Values are presented as mean ± standard deviation. The mean difference is significant at the 0.05 level. HRM: high-resolution manometry; VP: velopharynx, TB: tongue base; UES: upper esophageal sphincter; S: seconds.

**Table 3 children-09-01987-t003:** Threshold values for each HRM parameter with sensitivity, specificity and predictive values for identifying inhalation risk.

HRM Variable	Optimal Threshold Value	AUC (95% CI)	Sensitivity (%) (95% CI)	Specificity (%) (95% CI)	+LR	−LR
VP *p* max	155	1.00 (0.928–1.00)	100 (86.2–100)	100 (86.2–100)		0.00
VP s	0.76	0.86 (0.734–0.943)	84 (63.9–95.4)	76 (50.6–87.9)	3.50	0.21
TB *p* max	205	0.970 (0.877–0.996)	100 (86.2–100)	88 (68.8–97.3)	8.33	0.00
TB s	0.52	0.811 (0.675–0.908)	72 (50.6–87.9)	76 (54.9–90.6)	3.00	0.37
UES *p* max	285	0.707 (0.561–0.827)	80 (59.3–93.1)	60 (38.7–78.8)	2.00	0.33
UES *p* min	4.3	0.931 (0.822–0.983)	80 (59.3–93.1)	92 (73.9–98.8)	10.00	0.22
UES relax	0.6	0.988 (0.906–0.995)	100 (86.2–100)	92 (73.9–98.8)	12.5	0.00
UES rest	65	1.00 (0.928–1.00)	100 (86.2–100)	100 (86.2–100)		0.00

HRM: high-resolution manometry; CI: confidence interval; LR: likelihood ratio; AUC: area under ROC curve; VP: velopharynx; TB: tongue base; UES: upper esophageal sphincter; *p*: pressure; s: seconds; max: maximal; min: minimal; rest: resting; relax: relaxation.

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
