# Peer review of "Evaluation of Dysphagia and Inhalation Risk in Neurologically Impaired Children Using Esophageal High-Resolution Manometry with Swallowing Analysis"

_children, 2022, doi:10.3390/children9121987_

Round 1

Reviewer 1 Report

Line 33: Please remove the reference to your study being the first of its kind here, or you should explain what this is based on.

Line 65-66: Please explain where this is based.

Line 73-83: While it is stated that there is a control group, no mention is made of this and the clear criteria for entry and exclusion of participants.

The same applies to patients no mention is made of the entry and exclusion criteria of the patients as well as no mention of the sampling method followed should be done to help the reader. Please make the required improvements.

Line 255-263: I feel that the conclusion paragraph needs improvement to summarize the most important findings of your study even if this is a first approach to the topic. Please make the required improvements.

General comment please include study limitations.

Author Response

Thank you for the positive comments on English and style.  We really appreciated your comments and followed your suggestion as specified in the point-by-point reply:

  • We modified the sentence in line 33 removing the reference to “our first study”
  • We modified the sentence in line 65-66: the aspiration’s definition that we have considered, is based on reference of North America Committee related to “aspiration related illnesses” and on ESPGHAN WG in order to evaluate the indication to exclusive enteral feeding.
  • We better specified in line 73 that in the Parks’s study differences between two groups were founded but we decided not to specify in the text the detailed inclusion’s characteristics of their patients. We were strongly inspired by this study regarding the manometric described parameters but this study has included adult population comparing healthy subject (not applicable in pediatric studies) and patients with pharyngeal dysphagia followed in a rehabilitation unit (inclusion criteria: any clinical symptom, confirmed diagnosis of dysphagia by VFSS evaluation; exclusion of patients with severe psychiatric impairment or poor cooperation). We used different inclusion criteria because our study was aimed to neurological impaired children.
  • We better specified exclusion criteria in our groups
  • We better specified in the conclusion section the important findings of our study
  • We added study limitations

Reviewer 2 Report

I think this is a very interesting study, especially relevant for severely bedridden neurological patients.

However, there are a few remarks

- it is desirable to place a list of all abbreviations before the beginning of the article.

- more clearly describe patients in the control group.

- determine the correlation between VFSS and manometry and show it separately - this is very important

Author Response

Thank you for your comments; we appreciated and followed your suggestions and modified our paper in accordance with them, as specified in the point-by-point reply. We have highlighted the changes made in yellow in the text

  • We added a list of abbreviations at the end of paper
  • We better described patients of control group. We decided to exclude patients with esophageal malformation in order to avoid bias related to intrinsic esophageal motility as in esophageal atresia. This last group of patients is the subject of another study that we are conducting.
  • The better specified in the text the correlation between VFSS and HRM adding swallow sequence showed with VFSS in patients with cerebral palsy in order to better clarify and correlate the results obtained with HRM